# Enantioconvergent construction of stereogenic silicon via Lewis base-catalyzed dynamic kinetic silyletherification of racemic chlorosilanes

Tianbao Hu[1], Chen Zhao[1], Yan Zhang[2], Yuzhong Kuang[3], Lu Gao[1], Wanshu Wang[1], Zhishan Su [2] ✉ & Zhenlei Song [1] ✉

Organosilanes possessing an enantioenriched stereogenic silicon center are important in many branches of chemistry, yet they remain challenging to synthesize in a practical and scalable way. Here we report a dynamic kinetic silyletherification process of racemic chlorosilanes with (S)-lactates using 4-aminopyridine as a Lewis base catalyst. This enantioconvergent approach asymmetrically constructs the stereogenic silicon center in a different manner from traditional resolution or desymmetrization. A range of silylethers have been prepared with high diastereoselectivity on up to 10 g-scale, allowing the practical synthesis of diverse enantioenriched organosilane analogs.

Silicon lies vertically below carbon in Group 14. Like carbon, silicon can also bond with other groups tetravalently, and it exhibits chirality when the four substituents are different. Organosilanes with an enantioenriched stereogenic silicon center[1–9] are attracting growing interests not only for understanding chirality and chemistry beyond carbon, but also for preparing advanced materials (**I**[10] and **II**[11]), bioactive molecules (**III**[12]), probes in mechanistic studies (**IV**[13]) as well as chiral auxiliaries (**V**[14]), ligands (**VI**[15–17]) and others types of chiral reagents (**VII**[18,19]) in asymmetric transformations (Fig. 1a). While stereogenic carbon centers are ubiquitous in nature and can be accessed by an abundance of synthetic methods, the stereogenic silicon center is unnatural and much more synthetically challenging.

In fact, only two strategies have so far been widely used to asymmetrically construct enantioenriched stereogenic silicon centers (Fig. 1b). The first strategy relies on the resolution of racemic organosilanes either through diastereomer formation and separation[20,21], or occasionally through kinetic resolution[22]. This strategy suffers from a major practical limitation that the enantioenriched organosilanes cannot be obtained in more than 50% yield. A general protocol is the use of (-)-menthol to resolve racemic chlorosilanes[21]. In this approach, two diastereomeric silyl ethers are formed in an essentially equimolar ratio, and must be separated by fractional crystallization or repeated flash chromatography. The second strategy is the desymmetrization of prochiral organosilanes bearing two enantiotropic Si–H[23–30], Si–O[31], Si–C[32–41] or Si–Cl[15] bonds. Most reactions are catalyzed by expensive palladium or rhodium catalysts, therefore limiting the scalability and practicality of the strategy. A third strategy: enantioconvergent transformation of racemic organosilanes, has rarely been considered, except for recent progress in a Rh-catalyzed dynamic kinetic asymmetric intramolecular hydrosilylation of alkynes with racemic hydrosilanes[42], and dynamic kinetic asymmetric transformation of racemic tetraorgano allyl silanes into silylethers using imidodiphosphorimidate (IDPi) catalysts[43].

We began by exploiting the fact that silicon centers, particularly those with strong Lewis acidity, are prone to Lewis base-assisted racemization involving highly reactive silicon species with unstable chirality at the silicon, such as pentacoordinate silicates[44–46]. As depicted in Fig. 1c, we hypothesized that if we could establish a rapid

[1]Key Laboratory of Drug-Targeting and Drug Delivery System of the Education Ministry and Sichuan Province, Sichuan Engineering Laboratory for Plant-Sourced Drug and Sichuan Research Center for Drug Precision Industrial Technology, West China School of Pharmacy, Sichuan University, 610041 Chengdu, China. [2]Key Laboratory of Green Chemistry and Technology, Ministry of Education, College of Chemistry, Sichuan University, 610064 Chengdu, China. [3]School of Pharmacy, China Pharmaceutical University, 639 Longmian Avenue, 211198 Nanjing, Jiangsu, China. ✉ e-mail: suzhishan@scu.edu.cn; zhenleisong@scu.edu.cn

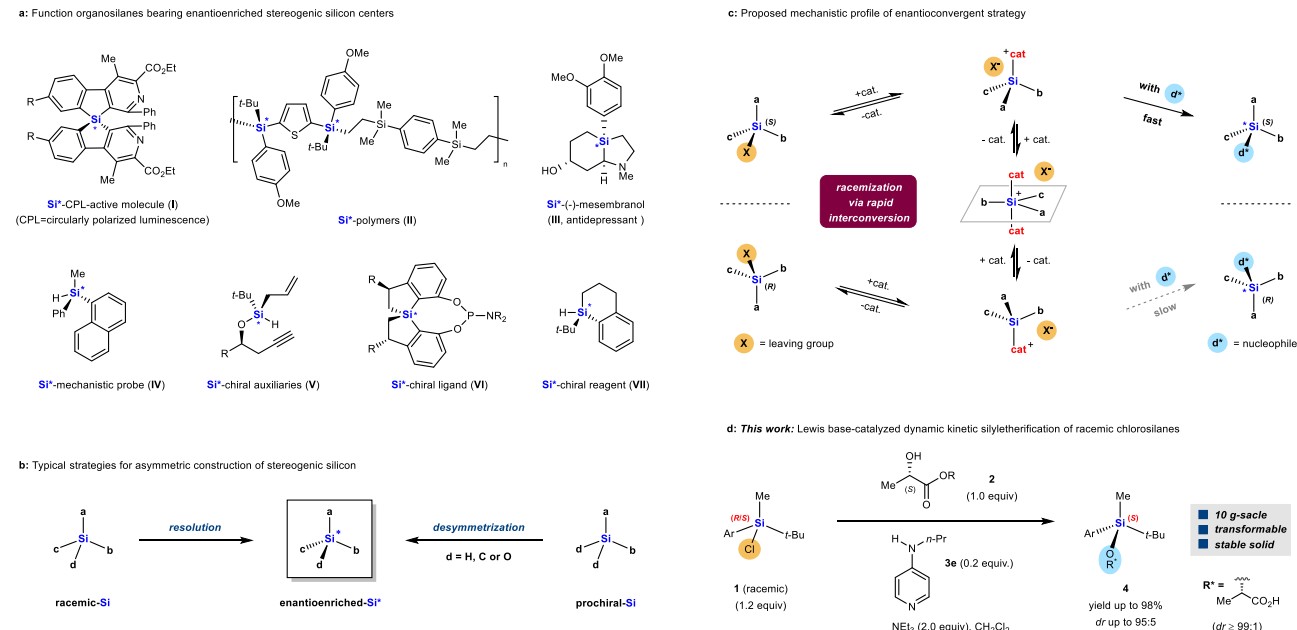

**Fig. 1 | Importance and synthesis of organosilanes bearing an enantioenriched stereogenic silicon center. a** Function organosilanes bearing enantioenriched stereogenic silicon centers. **b** Typical strategies for asymmetric construction of stereogenic silicon via resolution or desymmetrization. **c** Proposed mechanistic profile of enantioconvergent conversion of racemic organosilanes. **d** This work: Lewis base (**3e**)-catalyzed dynamic kinetic silyletherification of racemic chlorosilanes **1** with (*S*)-lactates **2** to give silicon-stereogenic silylethers **4**. *dr*: diastereomeric ratio.

interconversion between two enantiomeric silicon species (SiabcX) through the activation of a Lewis basic catalyst and that if one of the enantiomers reacted with the enantioenriched reactant d* much faster than the other enantiomer did, the starting racemic mixture would fully collapse to a single diastereomer bearing an enantioenriched silicon center [e.g. (*S*)-Siabcd]. Such enantioconvergence would be mechanistically distinct from traditional resolution or desymmetrization.

Here, we validate our hypothesis by designing and achieving a Lewis base-catalyzed dynamic kinetic[47] silyletherification process of racemic chlorosilanes **1** with easily accessible (*S*)-lactate analogs **2** in the presence of 4-aminopyridine **3e** as catalyst (Fig. 1d). This approach allowed a practical synthesis of silylethers **4** with high diastereoselectivity and in high yields on up to 10 g-scale. We were also able to transform silylethers **4** into enantioenriched hydrosilanes, deuterosilanes and tetraorganosilanes, which are difficult to synthesize using previously reported methods.

## Results

### Screening of reaction conditions

A variety of commercially available enantioenriched secondary alcohols were screened in CH$_2$Cl$_2$ at −78 °C using 1.2 equiv. of **1a**[21] as the model chlorosilane. No desired silylether **4a** was obtained using (*R*)−1-phenylethan-1-ol **2a** in the presence of 2.0 equiv. of NEt$_3$ or 2,6-lutidine, suggesting that these two Lewis bases do not activate **1a** (Table 1, entry 1). In contrast, 4-DMAP (**3a**), which functioned as both catalyst and auxiliary base, enabled facile silyletherification of either **2a** or (*L*)-menthol (**2b**), providing **4a** and **4b** in high yields, but with a diastereomeric ratio (*dr*) of only 50:50 (entries 2 and 3). Using (*S*)−1-(pyridin-2-yl)ethan-1-ol **2c** increased *dr* to 67:33 (entry 4), while reaction of **1a** with (*R*)-dimethyl malate (**2d**), (*R*)-pantolactone (**2e**) or methyl (*S*)-mandelate (**2f**) led to the respective products **4d, 4e** and **4f** with respective *dr*s of 84:16, 86:14 and 89:11 (entries 5-7). The optimal skeleton of α-carbonyl-substituted alcohols turned out to be (*S*)-lactates (**2g-i**, entries 7-9), in which 5-trifluoromethyl)furan-substituted **2i** provide silylether **4i** in 95% yield with *dr* of 92:8 (entry 10).

We also screened various 4-aminopyridine catalysts. Pyridonaphthyridine **3b**, the most catalytically active 4-DMAP analog reported by Steglich[48], provided a slightly lower *dr* of 91:9 (entry 11), while the 4-primary amine-substituted pyridine **3c** lowered both yield (54%) and *dr* (89:11) (entry 12). Using 4-aminopyridines with a 2-substitution such as **3d** further reduced yield (29%) and *dr* (67:33) (entry 13), perhaps because the 2-substitution sterically inhibited the interaction of the nitrogen on pyridine ring with the silicon center in chlorosilane. Conversely, the 4-secondary amine-substituted pyridine **3e**[49] improved *dr* to 94:6 (entry 14). Lengthening the alkyl group on the 4-nitrogen (**3f**) or making it bulkier (**3g**) lowered either yield or diastereoselectivity (entries 15 and 16). Using 4-phenylaminopyridine **3h** led to **4i** in only 45% yield (entry 17), probably because the lone pair of electrons on the 4-nitrogen competitively delocalized into the phenyl ring, weakening the Lewis basicity of the pyridine nitrogen.

The combination of 0.2 equiv. of **3e** as the catalyst and 2.0 equiv. of NEt$_3$ as an auxiliary base also functioned well, giving **4i** in 95% yield with *dr* of 93:7 (entry 18). Thus, we selected **3e** as the optimal catalyst, which is an air-stable orange solid and can be prepared in two steps from *tert*-butyl pyridin-4-ylcarbamate on a 10-g scale in an overall yield of 85%, without the need for silica gel chromatography. Either increasing the reaction temperature to −20 °C or lowering the loading of **3e** to 0.1 equiv. reduced silyletherification efficiency (entries 19 and 20). Switching the auxiliary base from NEt$_3$ to the less basic and more sterically hindered 2,6-lutidine led to **4i** in only 18% yield with a moderate *dr* of 87:13 (entry 21). This result implies that the auxiliary base may act as more than just an acid scavenger to influence the diastereochemical outcome.

### Scope of racemic chlorosilanes 1

With the optimal reaction conditions in hand (Table 1, entry 18), the scope of racemic chlorosilanes **1** was examined using alcohol **2i** (Fig. 2). Replacement of the Me group at position-a on the silicon with slightly bulkier Et, *n*-Pr or *n*-Bu groups predominantly gave (*S, S*)-silylethers **4j-l**, albeit with lower *dr*s. Although the *dr*s increased with the decreasing bulkiness of alkyl substitution-a, we were unable to derive a simple rule

## Table 1 | Screening of reaction conditions[a]

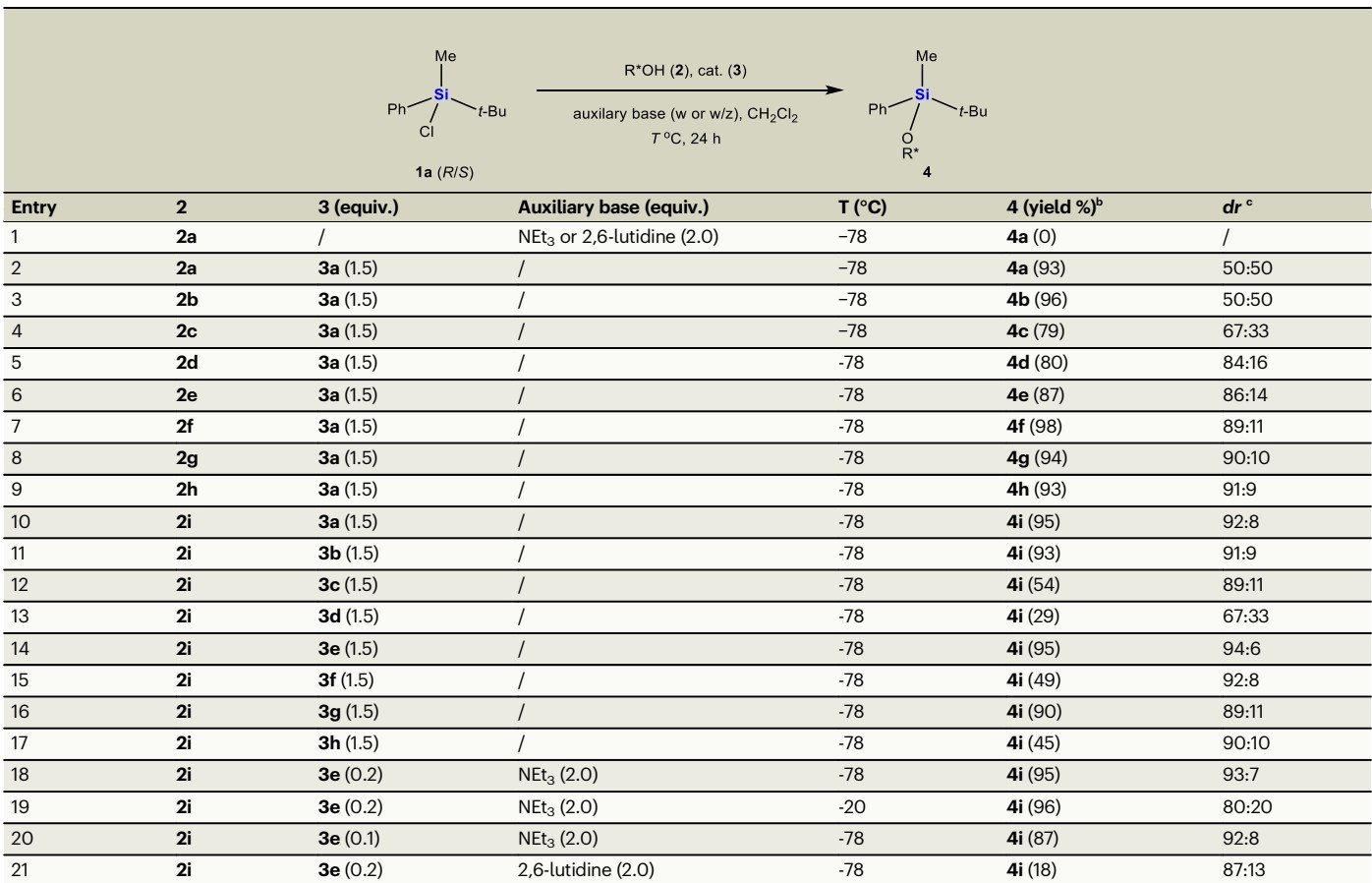

| Entry | 2 | 3 (equiv.) | Auxiliary base (equiv.) | T (°C) | 4 (yield %)[b] | dr [c] |
|---|---|---|---|---|---|---|
| 1 | 2a | / | NEt₃ or 2,6-lutidine (2.0) | −78 | 4a (0) | / |
| 2 | 2a | 3a (1.5) | / | −78 | 4a (93) | 50:50 |
| 3 | 2b | 3a (1.5) | / | −78 | 4b (96) | 50:50 |
| 4 | 2c | 3a (1.5) | / | −78 | 4c (79) | 67:33 |
| 5 | 2d | 3a (1.5) | / | -78 | 4d (80) | 84:16 |
| 6 | 2e | 3a (1.5) | / | -78 | 4e (87) | 86:14 |
| 7 | 2f | 3a (1.5) | / | -78 | 4f (98) | 89:11 |
| 8 | 2g | 3a (1.5) | / | -78 | 4g (94) | 90:10 |
| 9 | 2h | 3a (1.5) | / | -78 | 4h (93) | 91:9 |
| 10 | 2i | 3a (1.5) | / | -78 | 4i (95) | 92:8 |
| 11 | 2i | 3b (1.5) | / | -78 | 4i (93) | 91:9 |
| 12 | 2i | 3c (1.5) | / | -78 | 4i (54) | 89:11 |
| 13 | 2i | 3d (1.5) | / | -78 | 4i (29) | 67:33 |
| 14 | 2i | 3e (1.5) | / | -78 | 4i (95) | 94:6 |
| 15 | 2i | 3f (1.5) | / | -78 | 4i (49) | 92:8 |
| 16 | 2i | 3g (1.5) | / | -78 | 4i (90) | 89:11 |
| 17 | 2i | 3h (1.5) | / | -78 | 4i (45) | 90:10 |
| 18 | 2i | 3e (0.2) | NEt₃ (2.0) | -78 | 4i (95) | 93:7 |
| 19 | 2i | 3e (0.2) | NEt₃ (2.0) | -20 | 4i (96) | 80:20 |
| 20 | 2i | 3e (0.1) | NEt₃ (2.0) | -78 | 4i (87) | 92:8 |
| 21 | 2i | 3e (0.2) | 2,6-lutidine (2.0) | -78 | 4i (18) | 87:13 |

*dr* diastereomeric ratio

[a]General methods: *tert*-butylchloro(methyl)(phenyl)silane **1a** (0.12 mmol), secondary chiral alcohol **2** (0.1 mmol), 4-aminopyridine **3** (0.02 mmol) in CH₂Cl₂ (1.0 mL) at −78 °C for 24 h.

[b]Isolated yield.

[c]Determined by ¹H NMR.

that "smaller is better" at position-a: a sterically less demanding alkynyl group at that position (*A* value, 0.4 *vs* 1.7 for Me) led to **4m** with *dr* of only 50:50. The alkynyl group may be too small to provide diastereomeric differentiation. Replacing the Me group with the sterically least demanding H attenuated the electrophilicity of the silicon so much that silyletherification was completely inhibited to give **4n** even at room temperature. As for position-b on the silicon, replacing the *t*-Bu group with smaller *i*-Pr, *n*-Bu, vinyl groups or *p*-toluene group reduced *dr* to 50:50 (**4o-r**), while H again inhibited silyletherification (**4s**). These results indicate the importance of sufficient bulkiness at position-b for high diastereoselectivity.

In contrast to position-a and position-b, various aryl substitutions at position-c supported good diastereoselectivity. Phenyl rings containing a variety of 4-electron-donating-substituents provided

silylethers **4t-x** with generally high *dr*s. The reaction also functioned well to give silylethers **4y-ad** in which the silicon was bonded to phenyl rings with 3-mono-, 3,4- or 3,5-di-, or cyclobutane-fused substituents. In contrast, the steric characteristics of the 2-substitution on the phenyl ring influenced *dr*: small Me (**4ae**) or OMe groups (**4af**) slightly increased *dr* (94:6 *vs* 93:7 of **4i**), whereas a large phenyl group (**4ag**) lowered it to 72:28. Additional substitution at 3- or 5-positions disfavored formation of the (*S*, *S*)-diastereomer: *dr* was lower for **4ah** and **4ai** (88:12) than for **4ae** (94:6). Chlorosilanes with electron-withdrawing substituents on the phenyl ring served as good substrates, giving **4aj-ao**. The moderate *dr*s of **4al** (84:16) and **4am** (82:18) imply that an electron-deficient phenyl ring probably makes the chlorosilanes more reactive, reducing diastereomeric differentiation. Naphthalene, spirobifluorene and benzofuran on the silicon were tolerated, leading to

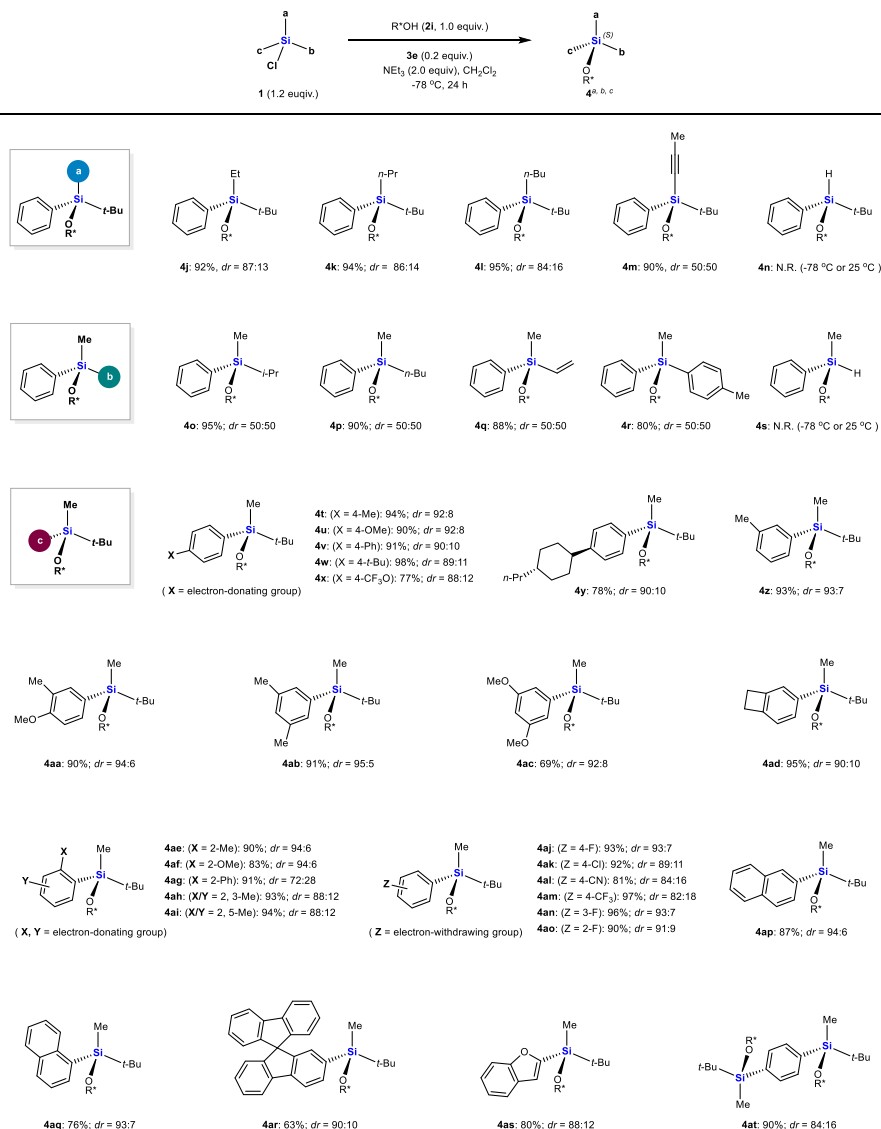

**Fig. 2 | Scope of racemic chlorosilanes 1.** [a]Method A for **4j-q** and **4 s: 2i** (0.2 mmol), **1** (0.24 mmol), **3e** (0.04 mmol), and NEt$_3$ (0.4 mmol) in CH$_2$Cl$_2$ (2.0 mL) at −78 °C for 24 h. Method B for **4r** and **4t-at**: the corresponding hydrosilane of **1** (0.3 mmol) and BPO (0.03 mmol) in CCl$_4$ (3 mL) at 100 °C for 24 h, followed by concentration and then reaction with **2i** (0.2 mmol), **3e** (20 mol %) and NEt$_3$ (0.4 mmol) in CH$_2$Cl$_2$ (2.0 mL) at −78 °C for 24 h. [b]Isolated yield. [c]Determined by [1]H NMR.

silylethers **4ap-as**. The reaction also functioned well for disilyletherification using phenyl-tethered bis(chlorosilane), leading to $C_2$-symmetric disilylether **4at** in 90% yield. The observed *dr* of 84:16 implies that the first generated stereogenic silicon center negligibly influences the stereochemistry of the second silyletherification.

## Mechanistic Studies

Analysis of reaction kinetics with different initial concentrations of components indicated first-order dependence on racemic chlorosilane **1a** and 4-aminopyridine catalyst **3e**, but saturation kinetics with secondary alcohol **2i** and NEt$_3$ (Fig. 3a)[50,51]. These findings suggest that the rate-limiting step in the overall reaction is the activation of chlorosilane **1a** by **3e**, after which the racemization of silicon and silylation with alcohol proceed rapidly.

Halosilanes are known to undergo Lewis base-catalyzed racemization[52–54]. One of the possible pathways has been proposed to involve interconversion of the ionic tetracoordinated silicon complex[55], which have been characterized or even isolated in the racemization or alcoholysis of halosilanes in the presence of pyridine[56],

imidazole[57] or NEt$_3$[58]. We hypothesize that such an intermediate may also account for the racemization between **Si$^S$-1a** and **Si$^R$-1a**.

Consistent with this hypothesis, [29]Si NMR spectra of the equimolar mixtures of **3e** with either *t*-BuPhMeSiBr **1a-Br** or *t*-BuPhMeSiOTf **1a-OTf** at room temperature in CD$_2$Cl$_2$ showed new, strong [29]Si resonances both appearing within ±0.3 ppm of 20.0 ppm (20.2 ppm for **1a-Br**/**3e** and 19.7 ppm for **1a-OTf**/**3e**) (Fig. 3b). Under the standard reaction conditions, reaction of **1a-Br** with **2i** provided **4i** in 96% yield with 93:7 *dr*, while reaction of **1a-OTf** with **2i** provided **4i** in 92% yield with 70:30 *dr*. These results support the existence of the ion-paired, tetracoordinated silicon species **5a-X**, in which the chemical shift of the silicon lies around 20.0 ppm regardless of the counterion. Mixing *t*-BuPhMeSiCl **1a-Cl** with **3e** at room temperature did not produce a new signal around 20.0 ppm, probably for thermodynamic reasons. In contrast, mixing the two compounds at −78 °C for 10 min led to complete disappearance of the **1a-Cl** signal at 25.1 ppm and appearance of a sharp peak at 19.7 ppm (Fig. 3c). These results suggest that formation of the ion-paired, tetracoordinated silicon intermediate **5a-Cl** is kinetically favored at −78 °C.

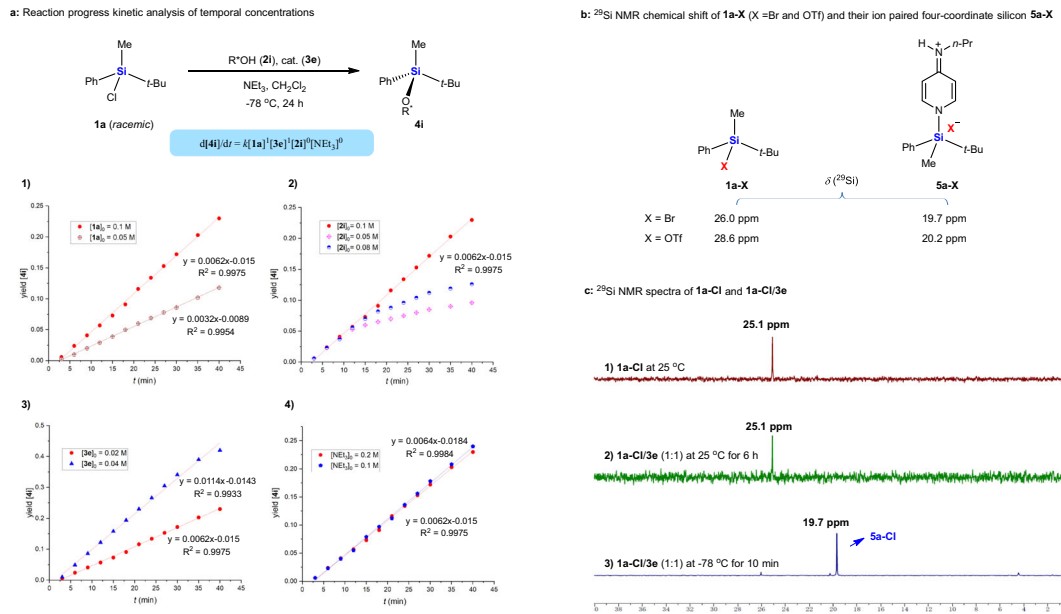

**Fig. 3 | Analysis of reaction kinetics and $^{29}$Si NMR studies. a** Reaction kinetics with different component concentrations. (1) First-order dependence on chlorosilane **1a**. (2) Saturation kinetics with secondary alcohol **2i**. (3) First-order dependence on 4-aminopyridine catalyst **3e**; (4) saturation kinetics with NEt₃. **b** $^{29}$Si NMR chemical shift of *t*-BuPhMeSiX **1a-X** (X = Br or OTf) and the ion-paired, tetracoordinated silicon **5a-X**, which was obtained by mixing **1a-X** with equimolar **3e** in CD₂Cl₂ at room temperature. **c** $^{29}$Si NMR spectra of **1a-Cl** and **1a-Cl/3e**. (1) **1a-Cl** in CD₂Cl₂ at room temperature. (2) Equimolar mixture of **1a-Cl** and **3e** in CD₂Cl₂ at room temperature. (3) Equimolar mixture of **1a-Cl** and **3e** in CD₂Cl₂ at −78 °C.

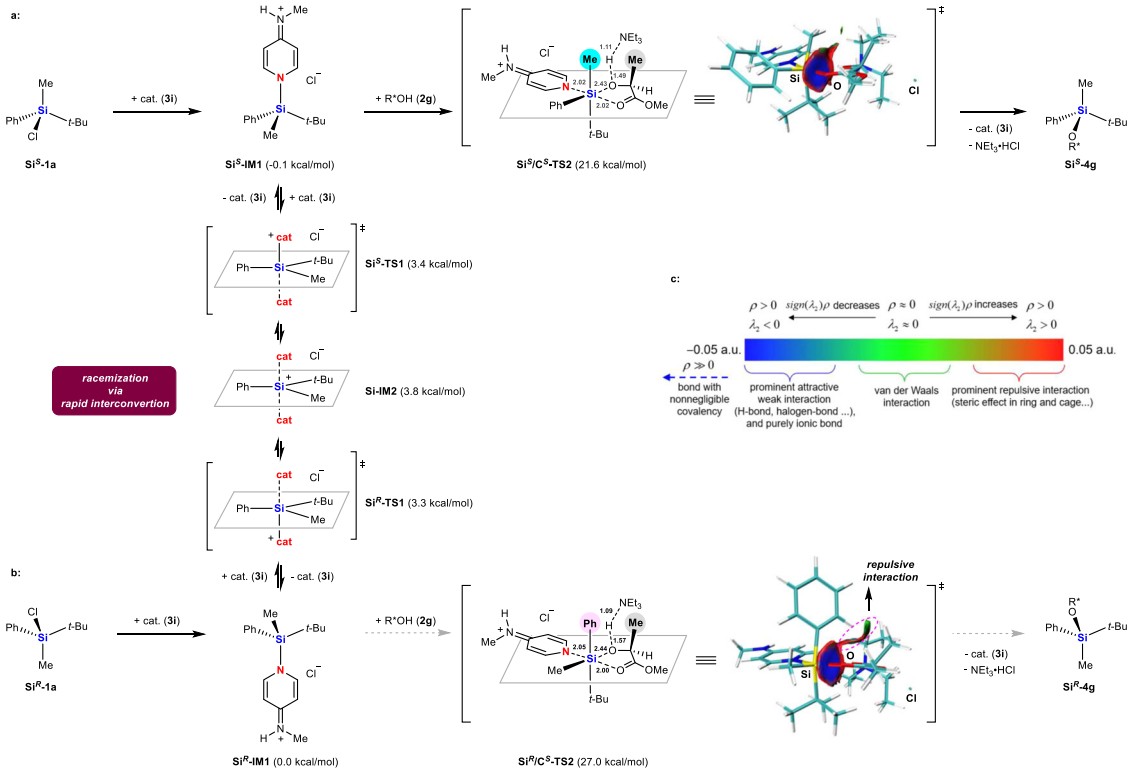

**Fig. 4 | Elucidation of the mechanism. a** **Si$^S$-1a** reacts with catalyst **3i** to give tetracoordinated silicon intermediate **Si$^S$-IM1**, which was transformed into **Si$^S$-4g** via the favorable transition state **Si$^S$/C$^S$-TS2** (21.6 kcal/mol); **b** **Si$^R$-1a** reacts with catalyst **3i** to give **Si$^R$-IM1**. Intermediate **Si$^R$-IM1** prefers to undergo racemization by rapid interconversion to give **Si$^S$-IM1** via pentacoordinate silicon **Si-IM2**, leading to **Si$^S$-4g** via **Si$^S$/C$^S$-TS2**, rather than reacts with **2g** to give **Si$^R$-4g** via the unfavorable transition state **Si$^R$/C$^S$-TS2** (27.0 kcal/mol); **c** c: Non-covalent interaction visualized by Multiwfn 3.8 (dev) software.

Based on these experimental results, we proposed the mechanism of our dynamic kinetic silyletherification of racemic chlorosilanes in Fig. 4. To simplify the calculation, we used *N*-methyl-substituted catalyst **3i**, which showed comparable efficiency with **3e**

to give **4g** in 95% yield with 91:9 *dr*. The n–σ* interaction between neutral **3i** and the racemic mixture **Si$^S$-1a** and **Si$^R$-1a** provides the ion-paired, tetracoordinated silicon intermediates **Si$^S$-IM1** and **Si$^R$-IM1**. This process is probably irreversible at −78 °C as suggested by the

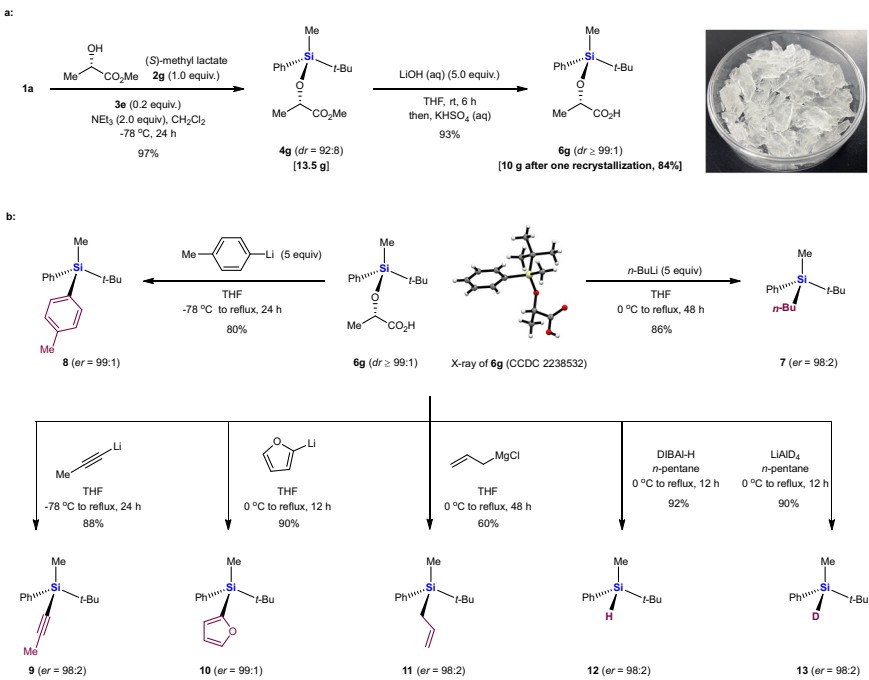

**Fig. 5 | Diverse transformations of the carboxylic acid analog 6g. a** 10-gram scale synthesis of **6g**; **b:** X-ray structure of **6g**, and transformations of **6g** into chiral organosilanes **7, 8, 9, 10, 11, 12,** and **13**.

[29]Si NMR spectra of the equimolar mixture of **1a** and **3i** (Fig. 3c). Even though the concentrations of **Si^S-IM1** and **Si^R-IM1** are low, the electrophilicity of their silicon is much greater than that in the neutral chlorosilane **1a** (*Lewis base activation of Lewis acid*[59]), facilitating the attack by the second **3i** molecule. DFT calculations predict that **Si^S-IM1** and **Si^R-IM1** interconvert through transition states **Si^S-TS1** (3.4 kcal/mol), **Si^R-TS1** (3.3 kcal/mol) and a common symmetrical pentacoordinate silicate intermediate **Si-IM2** (3.8 kcal/mol), leading to racemization of their silicon centers. Such an interconversion process is energetically very favorable, suggesting that it is rapid and reversible, consistent with the observed first-order dependence on catalyst in the overall reaction. The high electrophilicity of **Si^S-IM1** and **Si^R-IM1** also facilitates the subsequent irreversible silyletherification with (*S*)-lactates such as **2g**. DFT calculations suggest two hexacoordinate transition states **Si^S/C^S-TS2** and **Si^R/C^S-TS2** accounting for the formation of **Si^S-4g** and **Si^R-4g**, respectively. The O atoms of the ester group in **2g** coordinate to the Si atoms in the orientation *trans* to the catalyst, with the O...Si distances of 2.02 Å for **Si^S/C^S-TS2** and 2.00 Å for **Si^R/C^S-TS2**. Such activation promotes the nucleophilic attack of the hydroxy group in **2g** to silicon center, accompanied by deprotonation with NEt₃ and releasing the catalyst. In both cases, the most sterically demanding *t*-Bu group forces the chiral moiety in **2g** as far away from it as possible, while Me group acts as the least sterically demanding group, combining with the larger Ph group to produce the stereo-discrimination during formation of **Si^S-4g** and **Si^R-4g**.

DFT calculation predicts that **Si^S/C^S-TS2** (21.6 kcal/mol) is more stable than **Si^R/C^S-TS2** (27.0 kcal/mol) by 5.4 kcal/mol at 195 K, because the non-bonded repulsion between the Me group on the silicon and the α-Me group in **2g** in the case of **Si^S/C^S-TS2** appears to be less severe than that between the Ph group on the silicon and the α-Me group in **2g** in the case of **Si^R/C^S-TS2**, as suggested by the non-covalent interaction analysis. Thus, **Si^R-IM1** energetically prefers racemizing to **Si^S-IM1**, leading to the dynamic kinetic transformation of racemic **1a** into **Si^S-4g**, consistent with the observed experimental yield (97%) and *dr* (92:8) (Fig. 5).

## Diverse transformations

We confirmed the synthetic usefulness of our reaction by silylating racemic chlorosilane **1a** with (*S*)-methyl lactate **2g**, a commercially available chiral feed stock, leading to (*S*, *S*)-silylether **4g** on a 10-g scale, in 97% yield and *dr* of 92:8 (Fig. 5a). Treating **4g** with aqueous LiOH followed by one recrystallization provided the corresponding carboxylic acid analog **6g** as an air-stable colorless solid in 84% yield and *dr* ≥ 99:1. We were then able to convert **6g** into various enantioenriched organosilanes.

Reaction of **6g** with *n*-butyl-, phenyl, alkynyl or 2-furanyllithiums yielded tetraorganosilanes **7–10** in high yields, with retention of configuration[60–62] at silicon leading to *er*s up to 99:1 (Fig. 5b). Synthesis of **8** would be particularly challenging using typical resolution or desymmetrization strategies because the phenyl and 4-methyl phenyl rings are sterically similar. In addition to organolithiums, the Grignard reagent allyl magnesium chloride[63,64] served as a good nucleophile to afford synthetically useful chiral allylsilane **11** in 60% yield with *er* of 98:2. Reduction of **6g** with DIBAl-H[65,66] or LiAlD₄ generated, respectively, hydrosilane **12** in 92% yield or deuterosilane **13** in 90% yield, with retention of configuration at silicon in both cases (*er* = 98:2). Both **12** and **13** are promising chiral reductants, while deuterosilane **13** may also be useful for asymmetric deuteration in drug discovery[67–69].

## Discussion

We have developed an enantioconvergent synthesis of silicon-stereogenic silylethers that involves silyletherification of racemic chlorosilanes using (*S*)-lactates in the presence of 4-aminopyridine catalyst. Although the racemization of silicon poses a problem for asymmetric syntheses in other contexts, we have exploited it to achieve dynamic kinetic silyletherification process of racemic chlorosilanes. chlorosilane, whereas a wide range of aryl groups is tolerated. The rate-limiting step in the reaction appears to be n-σ* interaction between the catalyst and the silicon center in chlorosilane, tetra-coordinated silicon intermediate, which racemize via rapid inter-conversion assisted by the second catalyst molecule. The reaction provides scalable access to silylethers for subsequent preparation of

Article

diverse enantioenriched organosilanes, some of which are difficult to access by traditional synthetic methods.

Our approach may also provide the basis for Lewis base-catalyzed dynamic kinetic resolutions and dynamic kinetic asymmetric transformations involving stereogenic silicon centers. Those studies are currently underway in our group.

## Methods
### Method A
To a 10 mL round-bottom flask charged with **2i** (48 mg, 0.2 mmol, 1.0 equiv.) in dry CH$_2$Cl$_2$ (2 mL) were added chlorosilanes (0.24 mmol, 1.2 equiv.), **3e** (6 mg, 0.04 mmol, 20 mol%) and NEt$_3$ (56 $\mu$L, 0.4 mmol, 2.0 equiv.) under an inert atmosphere of argon at -78 °C. The mixture was stirred for 24 h at −78 °C before warming to room temperature and removing the solvent under reduced pressure. Purification by column chromatography on silica gel (gradient eluent: Petroleum Ether to Petroleum Ether/EtOAc = 30:1) afforded the desired product **4**.

### Method B
**Step 1**: To a 10 mL round-bottom flask charged with silanes (0.3 mmol) and CCl$_4$ (3 mL) was added benzoyl peroxide (8 mg, 0.03 mmol). The mixture was refluxed for 21 h followed by stirring for 3 h at room temperature before removing the solvent under reduced pressure. The crude product was used without purification in the next step.

**Step 2**: To a 10 mL round-bottom flask charged with **2i** (48 mg, 0.2 mmol, 1.0 equiv.) in dry CH$_2$Cl$_2$ (2 mL) were added the crude chlorosilane **1** (1.2 equiv.), **3e** (6 mg, 0.04 mmol, 20 mol%) and NEt$_3$ (56 $\mu$L, 0.4 mmol, 2.0 equiv.) under an inert atmosphere of argon at -78 °C. The mixture was stirred for 24 h at −78 °C before warming to room temperature and removing the solvent under reduced pressure. Purification by column chromatography on silica gel (gradient eluent: Petroleum Ether to Petroleum Ether /EtOAc = 3:1) afforded the desired product **4**.

## Data availability
All data generated in this study are provided in the Supplementary Information. The Cartesian coordinates are shown in the Supplementary Data 1. The source data of reaction kinetics are available and shown in the Supplementary Data 2. The X-ray crystallographic data used in this study are available in the Cambridge Crystallographic Data Center (CCDC) under accession code 2238532 (**6g**). Source data are provided with this paper.

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

## Acknowledgements

The authors are grateful to financial support from the National Natural Science Foundation of China (21921002, 22171191), National Key R&D Program of China (2022YFC2804202), Science and Technology Department of Sichuan Province (2020YFS0186), Science and Technology Major Project of Tibetan Autonomous Region of China (XZ202201ZD0001G), Sichuan University and Jiangsu Hengrui Pharmaceuticals West China Personnel Training and Discipline Development Funding (2018029) for financial support. We acknowledge Prof. P. C. Deng and Prof. D. B. Luo at Analytical Undefined Testing Center of Sichuan University for conducting $^{29}$Si NMR experiments (P.C.D.), X-ray crystallographic analysis of **6g** (D.B.L.).

## Author contributions

Z.L.S. and T.B.H. conceived and designed the project. Z.L.S. and Z.S.S. directed the project. T.B.H., C.Z., Y.Z.K. carried out the experiments. Z.S.S. and Y.Z. performed the DFT calculation. Z.L.S., Z.S.S., T.B.H., Y.Z., L.G., and W.S.W. prepared the manuscript. All authors analyzed the data and discussed the results.

## Competing interests

The authors declare no competing interests.
