## [Peer Review File · Nature Communications]

REVIEWER COMMENTS

Reviewer #1 (Remarks to the Author):

In this manuscript, Song and coworkers described a dynamic kinetic silyletherification process of racemic chlorosilanes with easily accessible or commercially available feed-stock chiral alcohols. Use of 4-aminopyridine Lewis base as the catalyst is the key to the success of this enantioconvergent transformation. A variety of tetrasubstituted Si-stereogenic silylethers have been synthesized in high yields with high diastereoselectivity. The substituents on silicon appear showing a wide scope, particularly the aryl substituents. Analysis of reaction kinetics, NMR studies of the intermediates, and DFT calculation provide deep and reasonable insight of the mechanism. What impresses me is the good practicality of the process shown in Figure 4. The resulting carboxylic acid-substituted silylether is an air-stable white solid and can be prepared from (S)-methyl lactate on 10 g-scale with $dr \geq 99:1$. This versatile platform molecule can be further transformed into a variety of functionalized chiral organosilanes. In contrast to recent extensive works focusing on construction of enantioenriched Si-stereogenic silane by desymmetrization of prochiral silicon, the enantioconvergent manner demonstrated in this work appears being novel. Overall, I believe that this is an excellent work and should be interesting to a wide scope of readership. I strongly support its publication in Nature Communications. I only have some very minor remarks as follow:

1. The bulky t-Bu group appears being crucial for the high dr. What about replace the t-Bu with SiMe₃ group?
2. How about chlorosilane containing two different aryl groups?
3. In the mechanistic studies, does the chiral alcohol has some interaction with catalyst? The NMR of their mixture might give some information.

Reviewer #2 (Remarks to the Author):

In this manuscript, Song and Su et al. developed a diastereoselective dynamic kinetic silyletherification of racemic chlorosilanes for the synthesis of tetrasubstituted Si-stereogenic silylethers. Although the construction of Si-stereogenic centers appears to be a topic of high current relevance in recent years, access to enantioenriched Si-stereogenic silane via enantioconvergent transformation of racemic organosilanes is rare (refs 48 and 49 by Xu and List). Key to the success of this work is the use of a 4-aminopyridine Lewis base catalyst, that delicately interacts with chlorosilane, allowing the racemization via the rapid interconversion of the tetracoordinated silicon

intermediate. Although a stoichiometric amount of chiral (S)-lactate was used in this process, given the concept of Lewis base-catalyzed enantioconvergent construction of stereogenic silicon, I think this manuscript should be considered to be accepted after the following issues are addressed by the authors.

1. The use of the specific secondary chiral alcohol featuring an ester group (lactate) is the key to achieve high dr, the authors should give a rational explanation.
2. The Me and tBu groups in the silane scaffold seem to be crucial in the transformation, which synergistically decides the stereocontrol of this process. Is that possible to rationalize the results? In addition, since the aromatic groups are competent in the reaction, how about silane substrates containing two different aryl groups in the transformation?
3. For the kinetic studies in Figure 2A, at least 4-5 different concentrations of each component should be conducted to give the order dependence accurately.
4. In Figure 2B, it is interesting to see the distinct behavior between 1a-Br, 1a-OTf, and 1a-Cl. I am curious what is the result treating of 1a-Br or 1a-OTf with ROH (2) under the standard reaction conditions.
5. Some recently published review papers should be cited, such as Synlett 2021, 32, 1281.

Reviewer #3 (Remarks to the Author):

Due to the instability of sp² hybridized Si atom, desymmetrization of prochiral silanes has emerged as the optimum strategy for the catalytic construction of Si-stereogenic centers during the past two decades. Recently dynamic kinetic transformation of racemic organosilanes offers an alternative in such case which has been disclosed by Xu (ref. 48) and List (ref. 49) starting from hydrosilanes and tetraorgano allyl silanes. In this manuscript Song, Su and coworkers reported a practical enantioconvergent approach to afford a wide array of silylethers in moderate to high diastereoselectivities through dynamic kinetic silyletherification of racemic chlorosilanes with chiral secondary alcohols catalyzed by easy-to-prepare pyridine derivatives. This work leverages the racemization of hypercovalent silicon intermediates via rapid interconversion to achieve dynamic kinetic silyletherification, which would serve as an effective complement to the current examples. In this context this reviewer recommends its publication in Nature Communications after the following issues are properly addressed:

(1) For Figure 1c, the authors aim to illustrate the enantioconvergent reaction scenario of the racemate, however the interconversion of the two corresponding intermediates seems confusing (middle part). The hypercovalent silicon species was suggested to add to the profile as depicted in Figure 3.

(2) For Table 2, variations of substituents at position a & b suffered decreased diastereoselectivities which to some extent limited the substrate scope. Have the authors attempted to modify the reaction parameters to further improve the outcome?

(3) For the mechanism studies, the rate-limiting step was determined to be the interaction between chlorosilane and Lewis base catalyst. Whereas DFT calculations indicated the energy barrier lied in the SN2 substitution of -OH group and deprotonation with Et3N which should be rate-determining. The authors should make a reasonable interpretation for the inconsistency.

(4) Some important papers are not cited and should be within the introduction part. Selected review: *Sci. China. Chem.* 2023, 66, doi.org/10.1007/s11426-022-1480-y Selected examples: *Angew. Chem., Int. Ed.* 2017, 56, 1125–1129; *Nat. Commun.* 2022, 13, 847. The citation of these papers would be informative for readers.

Dear Reviewers,

Thank you very much for giving the invaluable comments for the manuscript “**Enantioconvergent Construction of Stereogenic Silicon via Lewis Base-Catalyzed Dynamic Kinetic Silyletherification of Racemic Chlorosilanes**” (NCOMMS-23-13406A)

According to the Reviewers’ comments, we have made the detailed revisions. The following tables itemize point-by-point responses to these comments and highlight the changes we have made in the manuscript.

Thank you very much for your invaluable time and assistance!

Yours sincerely,

Prof. Zhenlei Song

Key Laboratory of Drug-Targeting of Education Ministry and Department of Medicinal Chemistry,
West China School of Pharmacy,
Sichuan University, Chengdu,
610041, P. R. China.

Response to Reviewer 1’s Comments

Comments	Responses
1. The bulky t -Bu group appears being crucial for the high dr . What about replace the t -Bu with SiMe ₃ group?	We tried the following process to synthesize chlorosilane C-1, which contains Ph, Me and SiMe₃ groups attached to the silicon center. However, the reaction only led to a complex mixture, and distillation of the crude product did not give the desired C-1. 1.0 equiv C-1
2. How about chlorosilane containing two different aryl groups?	We prepared chlorosilane S1ab, which contains Ph, Tol and Me groups attached to the silicon center. Under the optimal

reaction conditions, **S1ab** was transformed into silylether **4r** in 80% yield with 1:1 *dr*. The low diastereoselectivity indicates that Ph and Tol groups, which are sterically similar, cannot provide efficient diastereomeric differentiation. The above results have been added to Table 2, the corresponding text and supplementary materials.

3. In the mechanistic studies, does the chiral alcohol has some interaction with catalyst? The NMR of their mixture might give some information.

We mixed the chiral alcohol **2i** and catalyst **3e** (1:1 molar ratio) in CD_2Cl_2 at room temperature. The ^1H NMR and ^{19}F NMR did not show significant change compared with those of **2i**, indicating that there was no obvious interaction between **2i** and **3e**.

Response to Reviewer 2's Comments

Comments	Responses
1. The use of the specific secondary chiral alcohol featuring an ester group (lactate) is the key to achieve high dr , the authors should give a rational explanation.	According to this comment, we reperformed the DFT calculation (see the updated Figure 3 and the related text for more details.) DFT calculations suggest two hexacoordinate transition states Si^S/C^S-TS2 and Si^R/C^S-TS2 , in which the O atoms of the ester group in 2g coordinate to the Si atoms in the orientation trans to the catalyst, with the O...Si distances of 2.02 Å for Si^S/C^S-TS2 and 2.00 Å for Si^R/C^S-TS2 , as suggested by the non-covalent interaction analysis. DFT calculation predicts that Si^S/C^S-TS2 (21.6 kcal/mol) is more stable than Si^R/C^S-TS2 (27.0 kcal/mol) by 5.4 kcal/mol at 195 K, because the non-bonded repulsion between the Me group on the silicon and the α-Me group in 2g in the case of Si^S/C^S-TS2 appears to be less severe than that between the Ph group on the silicon and the α-Me group in 2g in the case of Si^R/C^S-TS2 . Thus, Si^R-IM1 energetically prefers racemizing to Si^S-IM1 , leading to the dynamic kinetic transformation of racemic 1a into Si^S-4g , consistent with the observed experimental yield (92%) and dr (92:8).
2. The Me and t -Bu groups in the silane scaffold seem to be crucial in the transformation, which synergistically decides the stereocontrol of this process. Is that possible to rationalize the results?	According to this comment, we reperformed the DFT calculation (see the updated Figure 3 and the related text for more details.) DFT calculations suggest two hexacoordinate transition states Si^S/C^S-TS2 and Si^R/C^S-TS2 accounting for the formation of Si^S-4g and Si^R-4g , respectively. In both cases, the most sterically demanding t -Bu group forces the chiral moiety in 2g as far away from it as possible, while Me group acts as the least sterically demanding group, combining with the larger Ph group to produce the stereo-discrimination during formation of Si^S-4g and Si^R-4g .
3. In addition, since the aromatic groups are competent in the reaction, how about silane substrates containing two different aryl groups in the transformation?	We prepared chlorosilane S1ab , which contains Ph, Tol and Me groups attached to the silicon center. Under the optimal reaction conditions, S1ab was transformed into silylether 4r in 80% yield with 1:1 dr . The low diastereoselectivity indicates that Ph and Tol groups, which have similar steric nature, cannot provide efficient diastereomeric differentiation. The above results have been added to Table 2, the corresponding text and supplementary materials.

	 $\text{S1ab} \xrightarrow[\text{NEt}_3 (2.0 \text{ equiv.}), \text{CH}_2\text{Cl}_2, 24 \text{ h}, -78 \text{ }^\circ\text{C}]{\text{2i}, \text{3e (0.2 equiv.)}}$ $\text{4r (yield = 80\%, dr = 1:1)}$ 
4. For the kinetic studies in Figure 2A, at least 4-5 different concentrations of each component should be conducted to give the order dependence accurately.	The kinetic studies were performed according to the following two important references: (1) Science, 2017, 356, 426-430; (2) Angew. Chem. Int. Ed. 2020, 59, 20814–20819. Both works reported a Lewis base-catalyzed dynamic kinetic asymmetric transformation for the enantioselective construction of stereogenic phosphorus (V) center, mechanistically similar to our enantioconvergent construction of stereogenic silicon via Lewis base-catalyzed dynamic kinetic silyletherification of racemic chlorosilanes. In these two works, reaction progress kinetic analysis of temporal concentration was carried out at 2-3 different concentrations for each component. We used such a method in our kinetic studies, and the resulting profiles shown in the Figure 2A appears accurate and reliable. We prefer to add the above two references (Science, 2017, 356, 426-430; Angew. Chem. Int. Ed. 2020, 59, 20814–20819.) in the text of “Mechanistic Studies” section as Ref. 50, Ref. 51.
5. In Figure 2B, it is interesting to see the distinct behavior between 1a-Br, 1a-OTf, and 1a-Cl. I am curious what is the result treating of 1a-Br or 1a-OTf with ROH (2) under the standard reaction conditions.	Under the standard reaction conditions, reaction of 1a-Br with 2i provided 4i in 96% yield with 93:7 dr, while reaction of 1a-OTf with 2i provided 4i in 92% yield with 70:30 dr. These results have been added as Ref. 59.
6. Some recently published review papers should be cited, such as Synlett 2021, 32, 1281.	This paper has been added as Ref. 9.

Response to Reviewer 3's Comments

Comments	Responses
1. For Figure 1c, the authors aim to illustrate the enantioconvergent reaction scenario of the racemate, however the interconversion of the two corresponding intermediates seems confusing (middle part). The hypercovalent silicon species was suggested to add to the	Figure 1c has been revised according to this comment. The achiral hypercovalent silicon species, which accounts for racemization, is shown in the middle part.

profile as depicted in Figure 3.	
2. For Table 2, variations of substituents at position a & b suffered decreased diastereoselectivities which to some extent limited the substrate scope. Have the authors attempted to modify the reaction parameters to further improve the outcome?	In this reaction, the combination of least sterically demanding Me group (a position) and most sterically demanding t-Bu group (b position) appears being essential for the high diastereoselectivity. For the combinations of a and b groups with less steric difference, chiral Lewis base catalysis might be a promising manner to give the high diastereoselectivity. We are working on this idea currently.
3. For the mechanism studies, the rate-limiting step was determined to be the interaction between chlorosilane and Lewis base catalyst. Whereas DFT calculations indicated the energy barrier lied in the S_N2 substitution of -OH group and deprotonation with Et₃N which should be rate-determining. The authors should make a reasonable interpretation for the inconsistency.	The reaction proceeds by two steps. In the first step, chlorosilane reacts with catalyst to give the ion-paired, tetracoordinated silicon intermediates Si^S-IM1 and Si^R-IM1. According to the results from NMR studies (Figure 2b and 2c), Si^S-IM1 and Si^R-IM1 are stable at -78 °C, and their formation is irreversible, and is the rate-limiting step of the reaction. In the second step, Si^S-IM1 and Si^R-IM1 serve as the real substrates to undergo interconversion and silyletherification. Because the second step accounts for the diastereoselectivity, we performed the DFT calculation from Si^S-IM1 and Si^R-IM1 as starting specie in the energy profiles (updated Figure 3). In this step, silyletherification via transition state Si^S/C^S-TS2 is the rate-limiting and diastereoselectivity-determining step, as depicted by the energy barrier.
4. Some important papers are not cited and should be within the introduction part. Selected review: Sci. China. Chem. 2023, 66, doi.org/10.1007/s11426-022-1480-y Selected examples: Angew. Chem., Int. Ed. 2017, 56, 1125–1129; Nat. Commun. 2022, 13, 847.	These papers have been added as Ref. 29, Ref. 34, Ref. 39.

REVIEWERS' COMMENTS

Reviewer #1 (Remarks to the Author):

The authors addressed all of my concerned questions. I recommend it for publication without further delay.

Reviewer #2 (Remarks to the Author):

The authors have addressed most of the questions properly asked by the referees. In addition, the authors give a reasonable explanation of the mechanism based on the experimental results and theoretical calculations. In summary, the authors develop a dynamic kinetic silyletherification process of racemic chlorosilanes with (S)-lactates using 4-aminopyridine as a Lewis base catalyst, which expands a new avenue for synthesizing diverse enantioenriched organosilane analogs. Therefore, I would like to recommend this paper for publication in Nature Communications.